# Pairwise Choice Markov Chains

**Stephen Ragain**
Management Science & Engineering
Stanford University
Stanford, CA 94305
sragain@stanford.edu

**Johan Ugander**
Management Science & Engineering
Stanford University
Stanford, CA 94305
jugander@stanford.edu

## Abstract

As datasets capturing human choices grow in richness and scale—particularly in online domains—there is an increasing need for choice models that escape traditional choice-theoretic axioms such as regularity, stochastic transitivity, and Luce's choice axiom. In this work we introduce the Pairwise Choice Markov Chain (PCMC) model of discrete choice, an inferentially tractable model that does not assume any of the above axioms while still satisfying the foundational axiom of *uniform expansion*, a considerably weaker assumption than Luce's choice axiom. We show that the PCMC model significantly outperforms both the Multinomial Logit (MNL) model and a mixed MNL (MMNL) model in prediction tasks on both synthetic and empirical datasets known to exhibit violations of Luce's axiom. Our analysis also synthesizes several recent observations connecting the Multinomial Logit model and Markov chains; the PCMC model retains the Multinomial Logit model as a special case.

## 1 Introduction

Discrete choice models describe and predict decisions between distinct alternatives. Traditional applications include consumer purchasing decisions, choices of schooling or employment, and commuter choices for modes of transportation among available options. Early models of probabilistic discrete choice, including the well known Thurstone Case V model [27] and Bradley-Terry-Luce (BTL) model [7], were developed and refined under diverse strict assumptions about human decision making. As complex individual choices become increasingly mediated by engineered and learned platforms—from online shopping to web browser clicking to interactions with recommendation systems—there is a pressing need for flexible models capable of describing and predicting nuanced choice behavior.

Luce's choice axiom, popularly known as the *independence of irrelevant alternatives* (IIA), is arguably the most storied assumption in choice theory [18]. The axiom consists of two statements, applied to each subset of alternatives $S$ within a broader universe $U$. Let $p_{aS} = \Pr(a \text{ chosen from } S)$ for any $S \subseteq U$, and in a slight abuse of notation let $p_{ab} = \Pr(a \text{ chosen from } \{a, b\})$ when there are only two elements. Luce's axiom is then that: (i) if $p_{ab} = 0$ then $p_{aS} = 0$ for all $S$ containing $a$ and $b$, (ii) the probability of choosing $a$ from $U$ conditioned on the choice lying in $S$ is equal to $p_{aS}$.

The BTL model, which defines $p_{ab} = \gamma_a/(\gamma_a + \gamma_b)$ for latent "quality" parameters $\gamma_i > 0$, satisfies the axiom while Thurstone's Case V model does not [1]. Soon after its introduction, the BTL model was generalized from pairwise choices to choices from larger sets [4]. The resulting Multinomal Logit (MNL) model again employs quality parameters $\gamma_i \geq 0$ for each $i \in U$ and defines $p_{iS}$, the probability of choosing $i$ from $S \subseteq U$, proportional to $\gamma_i$ for all $i \in S$. Any model that satisfies Luce's choice axiom is equivalent to some MNL model [19].

One consequence of Luce's choice axiom is *strict stochastic transitivity* between alternatives: if $p_{ab} \geq 0.5$ and $p_{bc} \geq 0.5$, then $p_{ac} \geq \max(p_{ab}, p_{bc})$. A possibly undesirable consequence of strict stochastic transitivity is the necessity of a total order across all elements. But note that strict

stochastic transitivity does not imply the choice axiom; Thurstone's model exhibits strict stochastic transitivity.

Many choice theorists and empiricists, including Luce, have noted that the choice axiom and stochastic transitivity are strong assumptions that do not hold for empirical choice data [9, 12, 13, 26, 28]. A range of discrete choice models striving to escape the confines of the choice axiom have emerged over the years. The most popular of these models have been Elimination by Aspects [29], mixed MNL (MMNL) [6], and nested MNL [22]. Inference is practically difficult for all three of these models [15, 23]. Additionally, Elimination by Aspects and the MMNL model also both exhibit the rigid property of regularity, defined below.

A broad, important class of models in the study of discrete choice is the class of *random utility models* (RUMs) [4, 20]. A RUM affiliates with each $i \in U$ a random variable $X_i$ and defines for each subset $S \subseteq U$ the probability $\Pr(i \text{ chosen from } S) = \Pr(X_i \geq X_j, \forall j \in S)$. An *independent RUM* has independent $X_i$. RUMs assume neither choice axiom nor stochastic transitivity. Thurstone's Case V model and the BTL model are both independent RUMs; the Elimination by Aspects and MMNL models are both RUMs. A major result by McFadden and Train establishes that for any RUM there exists a MMNL model that can approximate the choice probabilities of that RUM to within an arbitrary error [23], a strong result about the generality of MMNL models. The nested MNL model, meanwhile, is not a RUM.

Although RUMs need not exhibit stochastic transitivity, they still exhibit the weaker property of *regularity*: for any choice sets $A$, $B$ where $A \subseteq B$, $p_{xA} \geq p_{xB}$. Regularity may at first seem intuitively pleasing, but it prevents models from expressing framing effects [12] and other empirical observations from modern behavior economics [28]. This rigidity motivates us to contribute a new model of discrete choice that escapes historically common assumptions while still furnishing enough structure to be inferentially tractable.

**The present work.** In this work we introduce a conceptually simple and inferentially tractable model of discrete choice that we call the PCMC model. The parameters of the PCMC model are the off-diagonal entries of a rate matrix $Q$ indexed by $U$. The PCMC model affiliates each subset $S$ of the alternatives with a continuous time Markov chain (CTMC) on $S$ with transition rate matrix $Q_S$, whose off-diagonal entries are entries of $Q$ indexed by pairs of items in $S$. The model defines $p_{iS}$, the selection probability of alternative $i \in S$, as the probability mass of alternative $i \in S$ of the stationary distribution of the CTMC on $S$.

The transition rates of these CTMCs can be interpreted as measures of preferences between pairs of alternatives. Special cases of the model use pairwise choice probabilities as transition rates, and as a result the PCMC model extends arbitrary models of pairwise choice to models of setwise choice. Indeed, we show that when the matrix $Q$ is parameterized with the pairwise selection probabilities of a BTL pairwise choice model, the PCMC model reduces to an MNL model. Recent parameterizations of non-transitive pairwise probabilities such as the Blade-Chest model [8] can be usefully employed to reduce the number of free parameters of the PCMC model.

Our PCMC model can be thought of as building upon the observation underlying the recently introduced Iterative Luce Spectral Ranking (I-LSR) procedure for efficiently finding the maximum likelihood estimate for parameters of MNL models [21]. The analysis of I-LSR is precisely analyzing a PCMC model in the special case where the matrix $Q$ has been parameterized by BTL. In that case the stationary distribution of the chain is found to satisfy the stationary conditions of the MNL likelihood function, establishing a strong connection between MNL models and Markov chains. The PCMC model generalizes that connection.

Other recent connections between the MNL model and Markov chains include the work on Rank-Centrality [24], which employs a discrete time Markov chain for inference in the place of I-LSR's continuous time chain, in the special case where all data are pairwise comparisons.

Separate recent work has contributed a different discrete time Markov chain model of "choice substitution" capable of approximating any RUM [3], a related problem but one with a strong focus on ordered preferences. Lastly, recent work by Kumar et al. explores conditions under which a probability distribution over discrete items can be expressed as the stationary distribution of a discrete time Markov chain with "score" functions similar to the "quality" parameters in an MNL model [17].

The PCMC model is not a RUM, and in general does not exhibit stochastic transitivity, regularity, or the choice axiom. We find that the PCMC model does, however, obey the lesser known but fundamental axiom of *uniform expansion*, a weakened version of Luce's choice axiom proposed by Yellott that implies the choice axiom for independent RUMs [30]. In this work we define a convenient structural property termed *contractibility*, for which uniform expansion is a special case, and we show that the PCMC model exhibits contractibility. Of the models mentioned above, only Elimination by Aspects exhibits uniform expansion without being an independent RUM. Elimination by Aspects obeys regularity, which the PCMC model does not; as such, the PCMC model is uniquely positioned in the literature of axiomatic discrete choice, minimally satisfying uniform expansion without the other aforementioned axioms.

After presenting the model and its properties, we investigate choice predictions from our model on two empirical choice datasets as well as diverse synthetic datasets. The empirical choice datasets concern transportation choices made on commuting and shopping trips in San Francisco. Inference on synthetic data shows that PCMC is competitive with MNL when Luce's choice axiom holds, while PCMC outperforms MNL when the axiom does not hold. More significantly, for both of the empirical datasets we find that a learned PCMC model predicts empirical choices significantly better than a learned MNL model.

## 2 The PCMC model

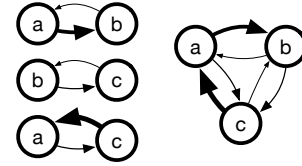

A Pairwise Choice Markov Chain (PCMC) model defines the selection probability $p_{iS}$, the probability of choosing $i$ from $S \subseteq U$, as the probability mass on alternative $i \in S$ of the stationary distribution of a continuous time Markov chain (CTMC) on the set of alternatives $S$. The model's parameters are the off-diagonal entries $q_{ij}$ of rate matrix $Q$ indexed by pairs of elements in $U$. See Figure 1 for a diagram. We impose the constraint $q_{ij} + q_{ji} \geq 1$ for all pairs $(i, j)$, which ensures irreducibility of the chain for all $S$.

Figure 1: Markov chains on choice sets $\{a, b\}$, $\{a, c\}$, and $\{b, c\}$, where line thicknesses denote transition rates. The chain on the choice set $\{a, b, c\}$ is assembled using the same rates.

Given a query set $S \subseteq U$, we construct $Q_S$ by restricting the rows and columns of $Q$ to elements in $S$ and setting $q_{ii} = -\sum_{j \in S \setminus i} q_{ij}$ for each $i \in S$. Let $\pi_S = \{\pi_S(i)\}_{i \in S}$ be the stationary distribution of the corresponding CTMC on $S$, and let $\pi_S(A) = \sum_{x \in A} \pi_S(x)$. We define the choice probability $p_{iS} := \pi_S(i)$, and now show that the PCMC model is well defined.

**Proposition 1.** *The choice probabilities $p_{iS}$ are well defined for all $i \in S$, all $S \subseteq U$ of a finite $U$.*

*Proof.* We need only to show that there is a single closed communicating class. Because $S$ is finite, there must be at least one closed communicating class. Suppose the chain had more than one closed communicating class and that $i \in S$ and $j \in S$ were in different closed communicating classes. But $q_{ij} + q_{ji} \geq 1$, so at least one of $q_{ij}$ and $q_{ji}$ is strictly positive and the chain can switch communicating classes through the transition with strictly positive rate, a contradiction. $\square$

While the support of $\pi_S$ is the single closed communicating class, $S$ may have transient states corresponding to alternatives with selection probability 0. Note that irreducibility argument needs only that $q_{ij} + q_{ji}$ be positive, not necessarily at least 1 as imposed in the model definition. One could simply constrain $q_{ij} + q_{ji} \geq \epsilon$ for some positive $\epsilon$. However, multiplying all entriesof $Q$ by some $c > 0$ does not affect the stationary distribution of the corresponding CTMC, so multiplication by $1/\epsilon$ gives a $Q$ with the same selection probabilities.

In the subsections that follow, we develop key properties of the model. We begin by showing how assigning $Q$ according a Bradley-Terry-Luce (BTL) pairwise model results in the PCMC model being equivalent to BTL's canonical extension, the Multinomial Logit (MNL) set-wise model. We then construct a $Q$ for which the PCMC model is neither regular nor a RUM.

### 2.1 Multinomial Logit from Bradley-Terry-Luce

We now observe that the Multinomial Logit (MNL) model, also called the Plackett-Luce model, is precisely a PCMC model with a matrix $Q$ consisting of pairwise BTL probabilities. Recall that the BTL model assumes the existence of latent "quality" parameters $\gamma_i > 0$ for $i \in U$ with $p_{ij} = \gamma_i/(\gamma_i + \gamma_j), \forall i, j \in U$ and that the MNL generalization defines $p_{iS} \propto \gamma_i, \forall i \in S$ for each $S \subseteq U$.

**Proposition 2.** *Let $\gamma$ be the parameters of a BTL model on U. For $q_{ji} = \frac{\gamma_i}{\gamma_i + \gamma_j}$, the PCMC probabilities $p_{iS}$ are consistent with an MNL model on $S$ with parameters $\gamma$.*

*Proof.* We aim to show that $\pi_S = \frac{\gamma}{||\gamma||_1}$ is a stationary distribution of the PCMC chain: $\pi_S^T Q_S = 0$. We have:

$$(\pi_S^T Q_S)_i = \frac{1}{||\gamma||_1} \left( \sum_{j \neq i} \gamma_j q_{ji} - \gamma_i (\sum_{j \neq i} q_{ji}) \right) = \frac{\gamma_i}{||\gamma||_1} \left( \sum_{j \neq i} \frac{\gamma_j}{\gamma_i + \gamma_j} - \sum_{j \neq i} \frac{\gamma_j}{\gamma_i + \gamma_j} \right) = 0, \; \forall i.$$

Thus $\pi_S$ is always the stationary distribution of the chain, and we know by Proposition 1 that it is unique. It follows that $p_{iS} \propto \gamma_i$ for all $i \in S$, as desired. □

Other parameterizations of $Q$, which can be used for parameter reduction or to extend arbitrary models for pairwise choice, are explored section 1 of the Supplementary material.

## 2.2 A counterexample to regularity

The regularity property stipulates that for any $S' \subset S$, the probability of selecting $a$ from $S'$ is at least the probability of selecting $a$ from $S$. All RUMs exhibit regularity because $S' \subseteq S$ implies $\Pr(X_i = \max_{j \in S'} X_j) \geq \Pr(X_i = \max_{j \in S} X_j)$. We now construct a simple PCMC model which does not exhibit regularity, and is thus not a RUM.

Consider $U = \{r, p, s\}$ corresponding to a rock-paper-scissors-like stochastic game where each pairwise matchup has the same win probability $\alpha > \frac{1}{2}$. Constructing a PCMC model where the transition rate from $i$ to $j$ is $\alpha$ if $j$ beats $i$ in rock-paper-scissors yields the rate matrix

$$Q = \begin{bmatrix} -1 & 1-\alpha & \alpha \\ \alpha & -1 & 1-\alpha \\ 1-\alpha & \alpha & -1 \end{bmatrix}.$$

We see that for pairs of objects, the PCMC model returns the same probabilities as the pairwise game, i.e. $p_{ij} = \alpha$ when $i$ beats $j$ in rock-paper-scissors, as $p_{ij} = q_{ji}$ when $q_{ij} + q_{ji} = 1$. Regardless of how the probability $\alpha$ is chosen, however, it is always the case that $p_{rU} = p_{pU} = p_{sU} = 1/3$. It follows that regularity does not hold for $\alpha > 2/3$.

We view the PCMC model's lack of regularity is a positive trait in the sense that empirical choice phenomena such as framing effects and asymmetric dominance violate regularity [12], and the PCMC model is rare in its ability to model such choices. Deriving necessary and sufficient conditions on $Q$ for a PCMC model to be a RUM, analogous to known characterization theorems for RUMs [10] and known sufficient conditions for nested MNL models to be RUMs [5], is an interesting open challenge.

## 3 Properties

While we have demonstrated already that the PCMC model avoids several restrictive properties that are often inconsistent with empirical choice data, we demonstrate in this section that the PCMC model still exhibits deep structure in the form of contractibility, which implies uniform expansion. Inspired by a thought experiment that was posed as an early challenge to the choice axiom, we define the property of contractibility to handle notions of similarity between elements. We demonstrate that the PCMC model exhibits contractibility, which gracefully handles this thought experiment.

## 3.1 Uniform expansion

Yellott [30] introduced *uniform expansion* as a weaker condition than Luce's choice axiom, but one that implies the choice axiom in the context of any independent RUM. Yellott posed the axiom of invariance to uniform expansion in the context of "copies" of elements which are "identical." In the context of our model, such copies would have identical transition rates to alternatives:

**Definition 1** (Copies). *For $i, j$ in $S \subseteq U$, we say that $i$ and $j$ are copies if for all $k \in S - i - j$, $q_{ik} = q_{jk}$ and $q_{ij} = q_{ji}$.*

Yellott's introduction to uniform expansion asks the reader to consider an offer of a choice of beverage from $k$ identical cups of coffee, $k$ identical cups of tea, and $k$ identical glasses of milk. Yellott contends that the probability the reader chooses a type of beverage (e.g. coffee) in this scenario should be the same as if they were only shown one cup of each beverage type, regardless of $k \geq 1$.

**Definition 2** (Uniform Expansion). *Consider a choice between $n$ elements in a set $S_1 = \{i_{11}, \ldots, i_{n1}\}$, and another choice from a set $S_k$ containing $k$ copies of each of the $n$ elements: $S_k = \{i_{11}, \ldots, i_{1k}, i_{21}, \ldots, i_{2k}, \ldots, i_{n1}, \ldots, i_{nk}\}$. The axiom of uniform expansion states that for each $m = 1, \ldots, n$ and all $k \geq 1$:*

$$p_{i_{m1}S_1} = \sum_{j=1}^{k} p_{i_{mj}S_k}.$$

We will show that the PCMC model always exhibits a more general property of contractibility, of which uniform expansion is a special case; it thus always exhibits uniform expansion.

Yellott showed that for any independent RUM with $|U| \geq 3$ the double-exponential distribution family is the only family of independent distributions that exhibit uniform expansion for all $k \geq 1$, and that Thurstone's model based on the Gaussian distribution family in particular does not exhibit uniform expansion.

While uniform expansion seems natural in many discrete choice contexts, it should be regarded with some skepticism in applications that model competitions. Sports matches or races are often modeled using RUMs, where the winner of a competition can be modeled as the competitor with the best draw from their random variable. If a competitor has a performance distribution with a heavy upper tail (so that their wins come from occasional "good days"), uniform expansion would not hold. This observation relates to recent work on team performance and selection [14], where non-invariance under uniform expansion plays a key role.

## 3.2 Contractibility

In a book review of Luce's early work on the choice axiom, Debreu [9] considers a hypothetical choice between three musical recordings: one of Beethoven's eighth symphony conducted by $X$, another of Beethoven's eighth symphony conducted by $Y$, and one of Debussy quartet conducted by $Z$. We will call these options $B_1$, $B_2$, and $D$ respectively. When compared to $D$, Debreu argues that $B_1$ and $B_2$ are indistinguishable in the sense that $p_{DB_1} = p_{DB_2}$. However, someone may prefer $B_1$ over $B_2$ in the sense that $p_{B_1B_2} > 0.5$. This is impossible under a BTL model, in which $p_{DB_1} = p_{DB_2}$ implies that $\gamma_{B_1} = \gamma_{B_2}$ and in turn $p_{B_1B_2} = 0.5$.

To address contexts in which elements compare identically to alternatives but not each other (e.g. $B_1$ and $B_2$), we introduce *contractible partitions* that group these similar alternatives into sets. We then show that when a PCMC model contains a contractible partition, the relative probabilities of selecting from one of these partitions is independent from how comparisons are made between alternatives in the same set. Our contractible partition definition can be viewed as akin to (but distinct from) *nests* in nested MNL models [22].

**Definition 3** (Contractible Partition). *A partition of $U$ into non-empty sets $A_1, \ldots, A_k$ is a contractible partition if $q_{a_i a_j} = \lambda_{ij}$ for all $a_i \in A_i, a_j \in A_j$ for some $\Lambda = \{\lambda_{ij}\}$ for $i, j \in \{1, \ldots, k\}$.*

**Proposition 3.** *For a given $\Lambda$, let $A_1, \ldots, A_k$ be a contractible partition for two PCMC models on $U$ represented by $Q, Q'$ with stationary distributions $\pi, \pi'$. Then for any $A_i$:*

$$\sum_{j \in A_i} p_{jU} = \sum_{j \in A_i} p'_{jU}, \tag{1}$$

*or equivalently, $\pi(A_i) = \pi'(A_i)$.*

*Proof.* Suppose $Q$ has contractible partition $A_1, \ldots, A_k$ with respect to $\Lambda$. If we decompose the balance equations (i.e. each row of $\pi^T Q = 0$), for $x \in A_1$ WLOG we obtain:

$$\pi(x) \left( \sum_{y \in A_1 \setminus x} q_{xy} + \sum_{i=2}^{k} \sum_{a_i \in A_i} q_{xa_i} \right) = \sum_{y \in A_1 \setminus x} \pi(y) q_{yx} + \sum_{i=2}^{k} \sum_{a_i \in A_i} \pi(a_i) q_{a_i x}. \tag{2}$$

Noting that for $a_i \in A_i$ and $a_j \in A_j$, $q_{a_i a_j} = \lambda_{ij}$, (2) can be rewritten:

$$\pi(x) \left( \sum_{y \in A_1 \setminus x} q_{xy} \right) + \pi(x) \sum_{i=2}^{k} |A_i| \lambda_{i1} = \sum_{y \in A_1 \setminus x} \pi(y) q_{yx} + \sum_{i=2}^{k} \pi(A_i) \lambda_{i1}.$$

Summing over $x \in A_1$ then gives

$$\sum_{x \in A_1} \pi(x) \left( \sum_{y \in A_1 \setminus x} q_{xy} \right) + \pi(A_1) \sum_{i=2}^{k} |A_i| \lambda_{i1} = \sum_{x \in A_1} \sum_{y \in A_1 \setminus x} \pi(y) q_{yx} + |A_1| \sum_{i=2}^{k} \pi(A_i) \lambda_{i1}.$$

The leftmost term of each side is equal, so we have

$$\pi(A_1) = \frac{|A_1| \sum_{i=2}^{k} \pi(A_i) \lambda_{i1}}{\sum_{i=2} |A_i| \lambda_{1i}}, \tag{3}$$

which makes $\pi(A_1)$ the solution to global balance equations for a different continuous time Markov chain with the states $\{A_1, \ldots, A_k\}$ and transition rate $\tilde{q}_{A_i A_j} = |A_j| \lambda_{ij}$ between state $A_i$ and $A_j$, and $\tilde{q}_{A_i A_i} = -\sum_{j \neq i} \tilde{q}_{A_i A_j}$. Now $q_{a_i a_j} + q_{a_j a_i} \geq 1$ implies $\lambda_{ij} + \lambda_{ji} \geq 1$. Combining this observation with $|A_i| > 0$ shows (as with the proof of Proposition 1) that this chain is irreducible and thus that $\{\pi(A_i)\}_{i=1}^{k}$ are well-defined. Furthermore, because $\tilde{Q}$ is determined entirely by $\Lambda$ and $|A_1|, \ldots, |A_k|$, we have that $\tilde{Q} = \tilde{Q}'$, and thus that $\pi(A_i) = \pi'(A_i), \forall i$ regardless of how $Q$ and $Q'$ may differ, completing the proof. $\square$

The intuition is that we can "contract" each $A_i$ to a single "type" because the probability of choosing an element of $A_i$ is independent of the pairwise probabilities between elements within the sets. The above proposition and the contractibility of a PCMC model on all uniformly expanded sets implies that all PCMC models exhibit uniform expansion.

**Proposition 4.** *Any PCMC model exhibits uniform expansion.*

*Proof.* We translate the problem of uniform expansion into the language of contractibility. Let $U_1$ be the universe of unique items $i_{11}, i_{21}, \ldots, i_{n1}$, and let $U_k$ be a universe containing $k$ copies of each item in $U_1$. Let $i_{mj}$ denote the $j$th copy of the $m$th item in $U_1$. Thus $U_k = \cup_{m=1}^{n} \cup_{j=1}^{k} i_{mj}$.

Let $Q$ be the transition rate matrix of the CTMC on $U_1$. We construct a contractible partition of $U_k$ into the $n$ sets, each containing the $k$ copies of some item in $U_1$. Thus $A_m = \cup_{j=1}^{k} i_{mj}$. By the definition of copies, that $\{A_m\}_{m=1}^{n}$ is a contractible partition of $U_k$ with $\Lambda = Q$. Noting $|A_m| = k$ for all $m$ in Equation (3) above results in $\{\pi(A_m)\}_{m=1}^{n}$ being the solution to $\pi^T Q = \pi^T \Lambda = 0$. Thus $p_{i_m U_1} = \pi(A_m) = \sum_{j=1}^{k} p_{i_{mj} U_k}$ for each $m$, showing that the model exhibits uniform expansion. $\square$

We end this section by noting that every PCMC model has a trivial contractible partition into singletons. Detection and exploitation of $Q$'s non-trivial contractible partitions (or appropriately defined "nearly contractible partitions") are interesting open research directions.

## 4  Inference and prediction

Our ultimate goal in formulating this model is to make predictions: using past choices from diverse subsets $S \subseteq U$ to predict future choices. In this section we first give the log-likelihood function $\log \mathcal{L}(Q; \mathcal{C})$ of the rate matrix $Q$ given a choice data collection of the form $\mathcal{C} = \{(i_k, S_k)\}_{k=1}^{n}$, where $i_k \in S_k$ was the item chosen from $S_k$. We then investigate the ability of a learned PCMC model to make choice predictions on empirical data, benchmarked against learned MNL and MMNL models, and interpret the inferred model parameters $\hat{Q}$. Let $C_{iS}(\mathcal{C}) = |\{(i_k, S_k) \in \mathcal{C} : i_k = i, S_k = S\}|$ denote the number of times in the data that $i$ was chosen out of set $S$ for each $S \subseteq U$, and let $C_S(\mathcal{C}) = |\{(i_k, S_k) \in \mathcal{C} : S_k = S\}|$ be the number of times that $S$ was the choice set for each $S \subseteq U$.

### 4.1 Maximum likelihood

For each $S \subseteq U$, $i \in S$, recall that $p_{iS}(Q)$ is the probability that $i$ is selected from set $S$ as a function of the rate matrix $Q$. After dropping all additive constants, the log-likelihood of $Q$ given the data $\mathcal{C}$ (derived from the probability mass function of the multinomial distribution) is:

$$\log \mathcal{L}(Q; \mathcal{C}) = \sum_{S \subseteq U} \sum_{i \in S} C_{iS}(\mathcal{C}) \log(p_{iS}(Q)).$$

Recall that for the PCMC model, $p_{iS}(Q) = \pi_S(i)$, where $\pi_S$ is the stationary distribution for a CTMC with rate matrix $Q_S$, i.e. $\pi_S^T Q_S = 0$ and $\sum_{i \in S} \pi_S(i) = 1$. There is no general closed form expression for $p_{iS}(Q)$. The implicit definition also makes it difficult to derive gradients for $\log \mathcal{L}$ with respect to the parameters $q_{ij}$. We employ SLSQP [25] to maximize $\log \mathcal{L}(Q; \mathcal{C})$, which is non-concave in general. For more information on the optimization techniques used in this section, see the Supplementary Materials.

### 4.2 Empirical data results

We evaluate our inference procedure on two empirical choice datasets, `SFwork` and `SFshop`, collected from a survey of transportation preferences around the San Francisco Bay Area [16]. The `SFshop` dataset contains 3,157 observations each consisting of a choice set of transportation alternatives available to individuals traveling to and returning from a shopping center, as well as a choice from that choice set. The `SFwork` dataset, meanwhile, contains 5,029 observations consisting of commuting options and the choice made on a given commute. Basic statistics describing the choice set sizes and the number of times each pair of alternatives appear in the same choice set appear in the Supplementary Materials[1].

We train our model on observations $T_{\text{train}} \subset \mathcal{C}$ and evaluate on a test set $T_{\text{test}} \subset \mathcal{C}$ via

$$\text{Error}(T_{\text{train}}; T_{\text{test}}) = \frac{1}{|T_{\text{test}}|} \sum_{(i,S) \in T_{\text{test}}} \sum_{j \in S} |p_{jS}(\hat{Q}(T_{\text{train}})) - \tilde{p}_{iS}(T_{\text{test}})|, \qquad (4)$$

where $\hat{Q}(T_{\text{train}})$ is the estimate for $Q$ obtained from the observations in $T_{\text{train}}$ and $\tilde{p}_{iS}(T_{\text{test}}) = C_{iS}(T_{\text{test}})/C_S(T_{\text{test}})$ is the empirical probability of $i$ was selected from $S$ among observations in $T_{\text{test}}$. Note that $\text{Error}(T_{\text{train}}; T_{\text{test}})$ is the expected $\ell_1$-norm of the difference between the empirical distribution and the inferred distribution on a choice set drawn uniformly at random from the observations in $T_{\text{test}}$. We applied small amounts of additive smoothing to each dataset.

We compare our PCMC model against both an MNL model trained using Iterative Luce Spectral Ranking (I-LSR) [21] and a more flexible MMNL model. We used a discrete mixture of $k$ MNL models (with $O(kn)$ parameters), choosing $k$ so that the MMNL model had strictly more parameters than the PCMC model on each data set. For details on how the MMNL model was trained, see the Supplementary Materials.

Figure 2 shows $\text{Error}(T_{\text{train}}; T_{\text{test}})$ on the `SFwork` data as the learning procedure is applied to increasing amounts of data. The results are averaged over 1,000 different permutations of the data with a 75/25 train/test split employed for each permutation. We show the error on the testing data as we train with increasing proportions of the training data. A similar figure for `SFshop` data appears in the Supplementary Materials.

We see that our model is better equipped to learn from and make predictions in both datasets, and when using all of the training data, we observe an error reduction of 36.2% and 46.5% compared to MNL and 24.4% and 31.7% compared to MMNL on `SFwork` and `SFshop` respectively.

Figure 2 also gives two different heat maps of $\hat{Q}$ for the `SFwork` data, showing the relative rates $\hat{q}_{ij}/\hat{q}_{ji}$ between pairs of items as well as how the total rate $\hat{q}_{ij} + \hat{q}_{ji}$ between pairs compares to total rates between other pairs. The index ordering of each matrix follows the estimated selection probabilities of the PCMC model on the full set of the alternatives for that dataset. The ordered options for `SFwork` are: (1) driving alone, (2) sharing a ride with one other person, (3) walking,

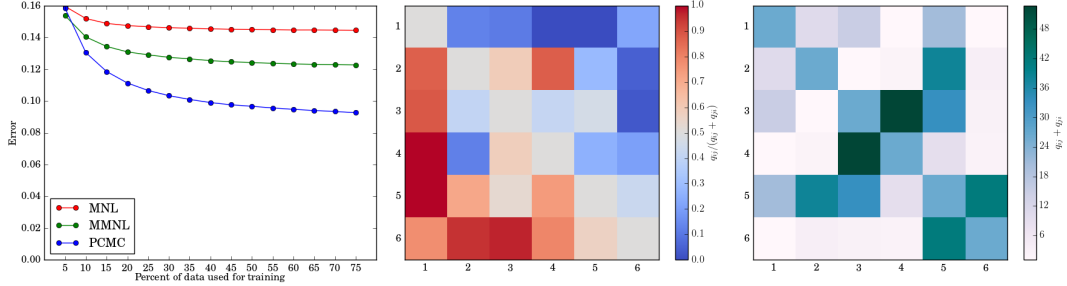

Figure 2: Prediction error on a 25% holdout of the `SFwork` data for the PCMC, MNL, and MMNL models. PCMC sees improvements of 35.9% and 24.5% in prediction error over MNL and MMNL, respectively, when training on 75% of the data.

(4) public transit, (5) biking, and (6) carpooling with at least two others. Numerical values for the entries of $\hat{Q}$ for both datasets appear in the Supplementary Materials.

The inferred pairwise selection probabilities are $\hat{p}_{ij} = \hat{q}_{ji}/(\hat{q}_{ji} + \hat{q}_{ij})$. Constructing a tournament graph on the alternatives where $(i, j) \in E$ if $\hat{p}_{ij} \geq 0.5$, cyclic triplets are then length-3 cycles in the tournament. A bound due to Harary and Moser [11] establishes that the maximum number of cyclic triples on a tournament graph on $n$ nodes is 8 when $n = 6$ and 20 when $n = 8$. According to our learned model, the choices exhibit 2 out of a maximum 8 cyclic triplets in the `SFwork` data and 6 out of a maximum 20 cyclic triplets for the `SFshop` data.

Additional evaluations of predictive performance across a range of synthetic datasets appear in the Supplementary Materials. The majority of datasets in the literature on discrete choice focus on pairwise comparisons or ranked lists, where lists inherently assume transitivity and the independence of irrelevant alternatives. The `SFwork` and `SFshop` datasets are rare examples of public datasets that genuinely study choices from sets larger than pairs.

## 5   Conclusion

We introduce a Pairwise Choice Markov Chain (PCMC) model of discrete choice which defines selection probabilities according to the stationary distributions of continuous time Markov chains on alternatives. The model parameters are the transition rates between pairs of alternatives.

In general the PCMC model is not a random utility model (RUM), and maintains broad flexibility by eschewing the implications of Luce's choice axiom, stochastic transitivity, and regularity. Despite this flexibility, we demonstrate that the PCMC model exhibits desirable structure by fulfilling uniform expansion, a property previously found only in the Multinomial Logit (MNL) model and the intractable Elimination by Aspects model.

We also introduce the notion of contractibility, a property motivated by thought experiments instrumental in moving choice theory beyond the choice axiom, for which Yellott's axiom of uniform expansion is a special case. Our work demonstrates that the PCMC model exhibits contractibility, which implies uniform expansion. We also showed that the PCMC model offers straightforward inference through maximum likelihood estimation, and that a learned PCMC model predicts empirical choice data with a significantly higher fidelity than both MNL and MMNL models.

The flexibility and tractability of the PCMC model opens up many compelling research directions. First, what necessary and sufficient conditions on the matrix $Q$ guarantee that a PCMC model is a RUM [10]? The efficacy of the PCMC model suggests exploring other effective parameterizations for $Q$, including developing inferential methods which exploit contractibility. There are also open computational questions, such as streamlining the likelihood maximization using gradients of the implicit function definitions. Very recently, learning results for nested MNL models have shown favorable query complexity under an oracle model [2], and a comparison of our PCMC model with these approaches to learning nested MNL models is important future work.

**Acknowledgements.**   This work was supported in part by a David Morgenthaler II Faculty Fellowship and a Dantzig–Lieberman Operations Research Fellowship.

## Footnotes

[1]Data and code available here: https://github.com/sragain/pcmc-nips

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
