[Supplementary Material]

# Pairwise Choice Markov Chains
# (Supplementary Material)

**Stephen Ragain**
Management Science & Engineering
Stanford University
Stanford, CA 94305
sragain@stanford.edu

**Johan Ugander**
Management Science & Engineering
Stanford University
Stanford, CA 94305
jugander@stanford.edu

## 1 Parameterizations of the $Q$ matrix

As observed in the main paper, if we parametrize $Q$ with pairwise probabilities from a BTL model with parameters $\gamma = \{\gamma_i\}_{i=1}^n$, i.e. $q_{ij} = p_{ji} = \frac{\gamma_j}{\gamma_i + \gamma_j}$, the resulting PCMC model is equivalent to an MNL model with parameters $\gamma$. In this section we explore some other ways of parameterizing $Q$ via a pairwise probability matrix $P$ with entries $p_{ij}$ and setting $Q = P^T$.

Blade-Chest models [2] are based on geometric $d$-dimensional embeddings of alternatives, where each alternative $i$ is parameterized with a blade vector $b_i \in \mathbb{R}^d$ and a chest vector $c_i \in \mathbb{R}^d$, in addition to a BTL-like quality parameter $\gamma_i > 0$. The Blade-Chest model comes in two variations: the *Blade-Chest distance model*, where

$$p_{ij}(b, c, \gamma) = S(||b_i - c_j||_2^2 - ||b_j - c_i||_2^2 + \gamma_i - \gamma_j),$$

where $S(x) = (1 + \exp(-x))^{-1}$ is the sigmoid/logistic function, and the *Blade-Chest inner product model*, where

$$p_{ij}(b, c, \gamma) = S(b_i \cdot c_j - b_j \cdot c_i + \gamma_i - \gamma_j).$$

The quality parameters $\gamma_i$ serve to connect the models to the BTL model, but do not meaningfully increase their expressiveness, so we disregard them in our use of the Blade-Chest model here. These two Blade-Chest models provide useful parameterizations of non-transitive pairwise probability matrices, $P(\theta)$, with $\theta = \{b, c\}$ consisting of the $2dn$ parameters of the "blade" and "chest" embeddings.

Another technique for parametrizing $Q$ involves representing $q_{ij}$ as a function of features of $i$ and $j$, i.e. $q_{ij} = f(X_i, X_j; \theta)$ where $X_i$ gives the salient features of $i$, and $\theta$ represents parameters that, for instance, give weights to these features. We can also formulate such analysis as a factoring $Q = W^T F - D$, where $W$ is a weight matrix, $F$ a feature matrix, and $D$ a diagonal matrix ensuring that the row sums of $Q$ are zero. We do not explore such parameterizations in this work, but merely highlight the potential to employ latent features of objects in a straight-forward manner, an approach closely related to conjoint analysis [4]. Such extensions would be similar to Chen and Joachims's work exploiting features of pairwise matchups by parametrizing the blades and chests as functions of those features [3].

## 2 Inference with synthetic data

We now evaluate our inference procedure's performance in three synthetic data regimes: (i) choice data generated from a PCMC model with $q_{ij}$ drawn i.i.d. uniformly from $[0, 1]$, (ii) choice data generated for a simple MNL model with qualities $\gamma$ drawn uniformly on the simplex, and (iii) choice data generated from a PCMC model with $Q$ parameterized by a two-dimensional Blade-Chest distance model. In order to create a strongly non-transitive instance of the Blade-Chest distance model, we draw the blades $b_i$ and chests $c_i$ uniformly at i.i.d. points along the two-dimensional unit circle, naturally producing many triadic impasses.

Figure 1: Learning error for PCMC, I-LSR, and Blade-Chest PCMC on synthetic data generated from (i) Arbitrary $Q$, (ii) MNL, and (iii) Blade-Chest models.

The PCMC model's $Q$ matrix has $n(n-1)$ parameters in general. When $Q$ is parameterized according to BTL we have just $n$ parameters, and when it is parameterized according to a Blade-Chest distance model in $d$ dimensions, we have $2dn$ parameters.

We evaluate each parameterization (arbitrary, MNL, Blade-Chest distance) for each synthetic regime. We employ the Iterative Luce Spectral Ranking (I-LSR) algorithm to learn the model under the BTL parameterization of $Q$, where the PCMC model is equivalent to an MNL model. When the data is generated by MNL, we expect MNL to outperform inference under the general parameterization. When the data is generated by a non-MNL PCMC model, we expect MNL to exhibit restricted performance compared to a general parameterization, since the data is not generated by a model that MNL can capture.

## 2.1 Synthetic data results

We generate training data $T_{\text{train}}$ and test data $T_{\text{test}}$ from each model using 25 randomly chosen triplets as choice sets. We then follow the inferential procedure in the main paper to evaluare the inferential efficacy of each of the three models on data generated according to each.

Figure 1 shows our error performance as the data grows, averaged across 10 instances, for each data generating process and each inference parameterization. We generate 5000 samples, assign 1000 of these to be testing samples, and incrementally add the other 4000 samples to the training data, tracking error on the testing samples as we increase the size of the training data set.

The inference is applied to a set $U$ with $n = 10$ objects, meaning that MNL has 20 parameters, the PCMC model with arbitrary $Q$ has $n(n-1) = 90$ parameters, and the PCMC model with the Blade-Chest distance parameterization in $\mathbb{R}^2$ uses $2dn = 40$ parameters. Overall we examine 9 data–model pairs, trained sequentially in 5 episodes, averaged across 10 instances. The figure thus represents 450 trained models.

As expected, the inferred PCMC models outperform MNL on data exhibiting IIA violations, while the MNL model learns the MNL data better, though PCMC is not far behind. More significantly, the Blade-Chest parametrization of the PCMC model performs very similarly to the general PCMC model in all three scenarios, despite having far fewer parameters. This is promising in domains where the $O(n^2)$ parameters of a general PCMC model is infeasible but a $O(n)$ parameterization using a Blade-Chest representation may be feasible.

## 3 Additional empirical data results and analysis

### 3.1 Optimization and smoothing

The MNL models were trained using I-LSR, a specialized algorithm for training Multinomial Logit models. Meanwhile, the PCMC likelihood was optimized using SQSLP [5] while the Mixed MNL models were trained using L-BFGS-B [1], which are both general purpose optimization algorithms available as part of the `scipy.optimize.minimize` software package. The reason for the different choices is that L-BFGS-B does not support the linear constraints that are part of the PCMC model likelihood. We choose to use L-BFGS-B for the MMNL model because it outperformed

Figure 2: Prediction error on SFshop data for the PCMC, MNL, and MMNL models. There are improvements of 31.3% and 19.2% in prediction error over MNL and MMNL respectively when training on 75% of the data.

SQSLP on SFtravel data, and we wanted to ensure that we were affording it the best possible chance to do well against the new model we contribute in this work.

The additive smoothing applied was $\alpha = 0.1$ for SFwork and $\alpha = 5$ for SFshop, where $\alpha$ is added to each $C_{iS}$ appearing in the likelihood function. The major motivator for the additive smoothing is that SQSLP occasionally goes awry on some of the permutations of the data, maintaining high error after a bad step. Even as currently formulated, the mean error improvement is somewhat underestimating the efficacy of the PCMC model, as a few bad runs out of 1000 will skew the distribution. Bad runs are easy to identify using cross-validation. Additionally, the implementation of SQSLP would try to compute numerical gradients through function evaluation at points outside the feasible space, which sometimes involved $q_{ij} + q_{ji} = 0$, violating the irreducibility of the chain. Additive smoothing prevents these issues at the cost of some efficacy of the model. We find the improvement in error of the PCMC model to be even more significant in light of the optimization issues involved, and find that it reinforces development of PCMC training algorithms as an interesting research area. Refining the optimization routines for learning PCMC models should be seen as important future work.

## 3.2 Empirical results for SFshop data

Figure 2 analyzes the SFshop dataset, repeating the analysis of SFwork found in the main paper. The indexing of $\hat{Q}$ is again according to the estimated selection probabilities on the full set of alternatives, which were: (1) drive alone both directions, (2) share a ride with one person in both directions, (3) share a ride with one person in one direction and drive alone in the other direction, (4) walk, (5) share a ride with more than one person in both directions, (6) share a ride with one person in one direction and more than one in the other direction, (7) bike, and (8) take public transit. The MMNL model here mixes $k = 6$ models, giving it 48 parameters while PCMC has 56 and MNL has 8. When using the full training set, PCMC performs 31.3% better than MNL and 19.2% better than MMNL.

## 3.3 Inferred $\hat{Q}$ matrices

The numerical values of the learned $\hat{Q}$ matrices, trained 100% of the data, are given below.

$$\hat{Q}_{\text{work}} = \begin{bmatrix} -3.875 & 2.314 & 0.557 & 0. & 0. & 1.004 \\ 18.17 & -29.571 & 0.776 & 1.836 & 2.075 & 6.713 \\ 4.84 & 7.752 & -35.994 & 1.042 & 14.476 & 7.884 \\ 1. & 0.105 & 0.456 & -13.147 & 3.65 & 7.937 \\ 21.201 & 9.108 & 3.323 & 7.363 & -47.7 & 6.704 \\ 11.459 & 3.014 & 0.117 & 5.67 & 12.334 & -32.594 \end{bmatrix}$$

$$\hat{Q}_{\text{shop}} = \begin{bmatrix} -35.264 & 1. & 0. & 1. & 0. & 0. & 5.142 & 28.122 \\ 0. & -12.959 & 3.363 & 0. & 0. & 2.03 & 2.433 & 5.133 \\ 1.635 & 0. & -22.945 & 0.637 & 0.243 & 0. & 4.877 & 15.553 \\ 0. & 12.73 & 5.95 & -24.455 & 2.174 & 0. & 1. & 2.601 \\ 1. & 3.487 & 4.458 & 0.194 & -15.366 & 0. & 5.227 & 1. \\ 1. & 1.143 & 5.788 & 6.841 & 6.344 & -31.747 & 6.15 & 4.482 \\ 1.331 & 1.305 & 0.136 & 0. & 0.226 & 0. & -30.693 & 27.695 \\ 0. & 0. & 0.402 & 10.521 & 0. & 0. & 1.602 & -12.526 \end{bmatrix}$$

## 3.4 Count matrices

Here we present data about the frequency with which pairs of alternatives appear in the same choice set. Matrices $A_{\text{work}}$ and $A_{\text{shop}}$ have as their $(i, j)$ entry the number of choice sets which contained both $i$ and $j$ in the SFwork and SFshop datasets respectively:

$$A_{\text{work}} = \begin{bmatrix} - & 1323 & 4755 & 3729 & 1658 & 4755 \\ 1323 & - & 1479 & 1395 & 797 & 1479 \\ 4755 & 1479 & - & 4003 & 1738 & 5029 \\ 3729 & 1395 & 4003 & - & 1611 & 4003 \\ 1658 & 797 & 1738 & 1611 & - & 1738 \\ 4755 & 1479 & 5029 & 4003 & 1738 & - \end{bmatrix}$$

$$A_{\text{shop}} = \begin{bmatrix} - & 3075 & 3075 & 3075 & 1844 & 3075 & 2069 & 2916 \\ 3075 & - & 3157 & 3157 & 1908 & 3157 & 2136 & 2997 \\ 3075 & 3157 & - & 3157 & 1908 & 3157 & 2136 & 2997 \\ 3075 & 3157 & 3157 & - & 1908 & 3157 & 2136 & 2997 \\ 1844 & 1908 & 1908 & 1908 & - & 1908 & 1908 & 1876 \\ 3075 & 3157 & 3157 & 3157 & 1908 & - & 2136 & 2997 \\ 2069 & 2136 & 2136 & 2136 & 1908 & 2136 & - & 2094 \\ 2916 & 2997 & 2997 & 2997 & 1876 & 2997 & 2094 & - \end{bmatrix}$$

The relative frequencies of choice sets of different sizes in the two datasets are given in Table 1.

Table 1:

| $|S| =$ | 3 | 4 | 5 | 6 | 7 | 8 |
|---|---|---|---|---|---|---|
| SFwork | 948 | 1918 | 1461 | 702 | - | - |
| SFshop | 0 | 1 | 131 | 902 | 311 | 1812 |