[Reviews · NeurIPS 2016]

Reviewer 1

Summary

This paper proposes a new, general pairwise ranking model, which satisfies the uniform expansion axiom. The paper claims that the model is a generalization of multinomial logit model, and is superior in prediction on two datasets.

Qualitative Assessment

The model is attractive. I have a small note: line 110: Let \pi_S be the stationary distribution... --> you need some assumptions to make sure that a/ a stationary distribution exists, and b/ it is unique. For instance, if the state-space is not connected, then \pi_S is not unique.

Confidence in this Review

2-Confident (read it all; understood it all reasonably well)


Reviewer 2

Summary

This paper proposes a new choice model, which satisfies contractability and does not exhibit rigid property of regularity: both of these properties have been suggested as desirable in explaining practical human choices. The model is based on continuous Markov chain, and builds upon recent advances in connecting choice models to Markov chains. This choice model is shown to explain real-world choice datasets better than the competing MNL model.

Qualitative Assessment

There are three desirable properties to choice models: 1) explaining real world human choices; 2) computational efficiency in learning the model from historical purchase data; 3) computational efficiency in solving optimization problems over those choice models in revenue management and assortment optimization. Typical choice models are on either side of two extremes. MNL model is efficient to learn and optimize, but often times fails to explain real datasets. On the other hands, Mixed MNL, Nested Logit, and other more complex models gain in descriptive and predictive power at the cost of significantly increased computational complexity both in learning and optimization. A recent advance in [3] has been an innovative attempt to bridge this gap, using Markov chain to define new choice models. The manuscript is in a similar line as [3], but addresses a very different aspect. The learning and optimization is still computationally hard for the proposed model, but it wins significantly in describing how human make choices. I believe that this is an innovative model, that perhaps is going to be studied by other researchers in this field of choice modeling and also be applied to solve real world problems.

Confidence in this Review

3-Expert (read the paper in detail, know the area, quite certain of my opinion)


Reviewer 3

Summary

The paper introduces Pairwise Choice Markov Chains model (PCMC) as a generalization of the Bradley-Terry-Luce model. PCMC doesn’t satisfy Luce’s choice axiom, regularity, or stochastic transitivity, but satisfies uniform expansion, a weaker condition than Luce’s choice axiom. Inference through maximum likelihood is proposed. Comparisons to multinomial logit model (MNL) on two real-world choice datasets show improvement over Iterative Luce Spectral Ranking (ILSR).

Qualitative Assessment

Despite being sometimes hard to follow, the proposed model is very interesting. The analysis of the model is satisfactory. My major concern is about its significance, and if NIPS is the right venue for this paper. The paper claims the violation of Luce’s choice axiom or regularity as a merit of the proposed model. I think it is debatable. The paper didn’t provide sufficient evidence to support this claim. The empirical results in Section 4 (PCMC vs. ILSR) don’t necessarily support to this claim. If uniform expansion is the desired property, can’t we directly work with paired probability distributions that satisfy it? It is mentioned that the inference through MLE is straightforward, but the application of BFGS is, at least to me, not that clear. The authors could elaborate. Line 22. Pab is not defined at this point. Line 45. Regularity is not defined at this point. It would be great to introduce framing effects. Line 237. With.

Confidence in this Review

2-Confident (read it all; understood it all reasonably well)


Reviewer 4

Summary

This paper introduces a new choice model which does not assume standard choice axioms like, Luce's axiom, stochastic transitivity and regularity. However, the model preserves uniform expansion property, rather a bit general property, contractibility introduced by the authors. Log-likelihood for this model is a non-concave function and, the optimal learning of the model parameters is not known. However, simulations show that model is more general than MNL model.

Qualitative Assessment

The newly introduced choice model seems promising with respect to the fact that it does not assume traditional choice axioms. However, I have some concerns. First, the log-likelihood of the model is non-concave function which poses difficulty in learning the model parameters. In simulations, authors learn the model parameters as a local optima of the log-likelihood function. Second, authors should compare the model prediction accuracy by comparing it with mixed MNL rather than MNL. It is natural that the proposed model would perform better than MNL as it has n^2 parameters whereas MNL has n parameters.

Confidence in this Review

2-Confident (read it all; understood it all reasonably well)


Reviewer 5

Summary

This paper considers the problem of developing flexible choice models that are not constrained to satisfy traditional, restrictive choice axioms (such as Luce’s axiom of independence of irrelevant attributes, IIA), but that can be tractably inferred from data. A (discrete) choice model over n items specifies probabilities of the form p(i,S) = Prob( i chosen from S ) for each subset of items S \subseteq [n] and each item i \in S. One of the most widely used models of discrete choice is the multinomial logit (MNL) choice model, which can be inferred efficiently from data but which is constrained to satisfy IIA and other restrictive assumptions. The paper proposes a new Markov chain based model of discrete choice that is parametrized by a (n x n) pairwise selection probability matrix. The model avoids several of the earlier restrictive assumptions, but is shown to satisfy an interesting property termed contractibility, which in turn also implies a reasonable property of uniform expansion. Parameter estimation in the model is done by maximum likelihood (the log-likelihood function is non-concave in general, but the experiments suggest that good parameters are learned). Previous models that avoid the above restrictive assumptions either do not satisfy even the uniform expansion property or are computationally difficult to learn, and so the proposed model offers unique properties in comparison to previous models. The proposed model also generalizes a recent Markov chain based formulation of the MNL model, which is subsumed as a special case. The proposed model is applied to two real data sets on transportation choices in the Bay Area, and in both cases, shows better predictive ability than the classical MNL model.

Qualitative Assessment

This is a strong paper, and additionally, is beautifully written and presented. It is one of the rare papers where I have very few comments or suggestions to make (just one question below). Truly beautiful work, and a pleasure to read! Ques: What is the running time of the proposed parameter estimation algorithm on the two data sets, and how does this compare with the running time for learning the MNL model? It would be nice to give details on this in the author rebuttal if possible, and to include them in the final version if accepted. In future work or a longer version, it would also be nice to include such comparisons with other models such as nested MNL etc. ---- After author response ----- I have read the author response and it addresses my question.

Confidence in this Review

3-Expert (read the paper in detail, know the area, quite certain of my opinion)


Reviewer 6

Summary

The paper describe a Markov model which outperforms the commonly used multinomial logistic model. The author(s) first present an overview of some of the earlier models (ie. Bradley-Terry-Luce, Thurstone models) which predates the multinomial logistic model and provides motivation of presenting a new model due to unrealistic assumptions (ie. transitivity amongst choices) that the early models hold. The paper presents a rich background into the basics regarding random utility models. The author(s) then present their contribution of the pair-wise choice Markov chain model, which is based on continuous time Markov chains, and provide the properties that their model holds and introduces a new property denoted as contractibility (referring to the grouping of similar choices). The author(s) perform experiments on two datasets (SFwork and SFshop) and provide empirical results compared to the basic multinomial logistic model on the task of predicting the mode of transportation for a commute in San Francisco.

Qualitative Assessment

The paper presents pair-wise choice Markov chain model (PCMC) as an alternative to the popular multinomial logistic model (MNL) coupled with proofs on properties that the model holds. This model can serve as an inference component to a number of methods and applications. The author(s) made a point that two alternatives may happen to be similar; however, the choice between choosing a third class versus each of these similar choices should yield different likelihoods. They then introduced the notion of contractibility. The scenario presented is in fact very common in many fields (ie. natural language processing) and mapping these alternatives to a common semantic class is not new. The experimental section of this paper appears to be weak. The experiments supposedly were performed on two datasets against multinomial logistic regression; however, the figure shows that it was matched against the Iterative Luce Spectral Ranking (I-LSR). Perhaps I-LSR was used to estimate the MNL model parameters? The empirical results show small improvements (from ~0.24 down to ~0.22, and ~0.17 to ~0.15, so ~0.02 on average error) over the MNL model. It would be interesting to see how the PCMC model compares to that was mentioned in some of the literature that was reviewed such as nested-MNL and Eliminations by Aspects. The aforementioned methods dates to the 1970s and 80s, perhaps a search for more recent literature and empirical comparisons amongst the methods can strengthen the paper.

Confidence in this Review

2-Confident (read it all; understood it all reasonably well)